# Herpes simplex virus 2 (HSV-2) evolves faster in cell culture than HSV-1 by generating greater genetic diversity

**Alberto Domingo López-Muñoz**[¤a], **Alberto Rastrojo**[¤b], **Rocío Martín**,
**Antonio Alcamí** *

Centro de Biología Molecular Severo Ochoa (Consejo Superior de Investigaciones Científicas and Universidad Autónoma de Madrid), Madrid, Spain

¤a Current address: Cellular Biology Section, Laboratory of Viral Diseases, NIAID, NIH, Bethesda, Maryland, United States of America
¤b Current address: Genetic Unit, Department of Biology, Universidad Autónoma de Madrid, Madrid, Spain
* aalcami@cbm.csic.es

**Data Availability Statement:** Raw sequence reads are available at the European Bioinformatics Institute (EMBL-EBI) European Nucleotide Archive

## Abstract

Herpes simplex virus type 1 and 2 (HSV-1 and HSV-2, respectively) are prevalent human pathogens of clinical relevance that establish long-life latency in the nervous system. They have been considered, along with the *Herpesviridae* family, to exhibit a low level of genetic diversity during viral replication. However, the high ability shown by these viruses to rapidly evolve under different selective pressures does not correlates with that presumed genetic stability. High-throughput sequencing has revealed that heterogeneous or plaque-purified populations of both serotypes contain a broad range of genetic diversity, in terms of number and frequency of minor genetic variants, both *in vivo* and *in vitro*. This is reminiscent of the quasispecies phenomenon traditionally associated with RNA viruses. Here, by plaque-purification of two selected viral clones of each viral subtype, we reduced the high level of genetic variability found in the original viral stocks, to more genetically homogeneous populations. After having deeply characterized the genetic diversity present in the purified viral clones as a high confidence baseline, we examined the generation of *de novo* genetic diversity under culture conditions. We found that both serotypes gradually increased the number of *de novo* minor variants, as well as their frequency, in two different cell types after just five and ten passages. Remarkably, HSV-2 populations displayed a much higher raise of nonconservative *de novo* minor variants than the HSV-1 counterparts. Most of these minor variants exhibited a very low frequency in the population, increasing their frequency over sequential passages. These new appeared minor variants largely impacted the coding diversity of HSV-2, and we found some genes more prone to harbor higher variability. These data show that herpesviruses generate *de novo* genetic diversity differentially under equal *in vitro* culture conditions. This might have contributed to the evolutionary divergence of HSV-1 and HSV-2 adapting to different anatomical niche, boosted by selective pressures found at each epithelial and neuronal tissue.

(ENA) as Bioproject ID PRJEB32133 and PRJEB32148.

**Funding:** This work was funded by Spanish Ministry of Science and Innovation and European Union (European Regional Development's Funds, FEDER) (grants SAF2015-67485-R and RTI2018-097581-B100) to AA. www.ciencia.gob.es/portal/site/MICINN/ A. D. L.-M. received funds from PhD studentship from Ministerio de Educación y Formación Profesional (FPU13/05425). www.educacionyfp.gob.es/portada.html The funders had no role in study design, data collection and analysis, decision to publish, or preparation of the manuscript.

**Competing interests:** The authors have declared that no competing interests exist.

## Author summary

Herpesviruses are highly human pathogens that establish latency in neurons of the peripheral nervous system. Colonization of nerve endings is required for herpes simplex virus (HSV) persistence and pathogenesis. HSV-1 global prevalence is much higher than HSV-2, in addition to their preferential tendency to infect the oronasal and genital areas, respectively. How these closely related viruses have been adapting and evolving to replicate and colonize these two different anatomical areas remains unclear. Herpesviruses were presumed to mutate much less than viruses with RNA genomes, due to the higher fidelity of the DNA polymerase and proofreading mechanisms when replicating. However, the worldwide accessibility and development of high-throughput sequencing technologies have revealed the heterogenicity and high diversity present in viral populations clinically isolated. Here we show that HSV-2 mutates much faster than HSV-1, when compared under similar and controlled cell culture conditions. This high mutation rate is translated into an increase in coding diversity, since the great majority of these new mutations lead to nonconservative changes in viral proteins. Understanding how herpesviruses differentially mutate under similar selective pressures is critical to prevent resistance to anti-viral drugs.

## Introduction

Herpes simplex virus (HSV) is well-known for being one of the most prevalent neurotropic pathogens worldwide, causing a broad range of diseases in humans. The latest epidemiological studies estimate around 66% and 13.2% of global seroprevalence for HSV-1 and HSV-2, respectively, depending on age, sex, and geographical region [1]. HSV-1 infects the oral mucosa preferentially, causing characteristic minor skin lesions and occasionally, encephalitis. Genital herpes is more often caused by HSV-2. Nevertheless, both viruses can infect either mucosa, frequently as a consequence of oral-genital sex [2–5]. After initial replication and dissemination at epithelial tissue, HSV infects sensory neuronal endings innervating the tissue [6,7]. Once the virus reaches the trigeminal ganglia or the dorsal root ganglia, the virus establishes latency for the host's lifetime. The virus periodically reactivates from latent to replicative stage, traveling anterogradely back to the epithelial tissue, where recurrent infection and transmission occur, exhibiting associated disease symptoms [8]. During these multiple cycles of latency and reactivation, the virus may reinfect the nervous system or be transmitted to a new host, finding numerous chances to expand its genetic repertoire for subsequent reactivation cycles [9,10]. The genetic diversity of the reactivating viral population can evolve by genetic drift, allowing the virus to respond to the host selective pressures that it faces when replicating. Other than the selective pressures due to immunological surveillance at the replication site, the virus may undergo selection in response to the differences between epithelial and neuronal environments [11–13]. HSV-2 is more similar to Chimpanzee herpes virus than it is to HSV-1, in terms of genomic sequence. Evolutionary studies have suggested that Chimpanzee herpes virus and one of the human HSV might have arisen via host–virus codivergence, as a result of a cross-species transmission event [14,15]. Since HSV-1 and HSV-2 exhibit a preference for infecting different anatomical areas, together with a hypothetically distant evolutionary origin, it is reasonable to think that the selective pressures during each cycle of latency and reactivation might have also contributed differentially to their evolution in humans, adapting their life cycle to each epithelial and neuronal niche.

The HSV genome is 152–155 Kilobase pairs long of double-stranded DNA, varying slightly between subtypes and strains. The genome is organized as unique long (UL) and unique short (US) coding regions, flanked by terminal/internal long/short structural repeats (i.e., T/I-Repeat(R)-L/S). The "a" sequence is present at the IRL-IRS border, but also at the termini of the TRL and TRS, enabling the inversion of the unique fragments orientation and producing four genomic isomers in equal ratios and functionality [16,17]. Genomic replication generates long concatemers of viral DNA, which are processed into unit-length genomes after cellular endonuclease G cleavage in the "a" sequence [18]. Those concatemers are highly branched, promoting recombination events between repeated regions and resulting in the inversion of the UL and US segments [19]. Interestingly, high recombination and inversion rates in the HSV genome have been extensively described, assisted by repetitive regions with a high G + C content [20–22]. Fluctuations in copy number, tandem repeats, and homopolymeric areas have been described as a frequent source of genetic variability in HSV-1 [23]. These mechanisms are critical components in the generation of variability within the large structural repeats of the HSV genome, which in fact contain immediate-early expressed genes essential to productive replication [8]. Nonetheless, genetic variability can be generated by different mechanisms other than recombination and copy number/length fluctuations, where genetic drift driven by polymerase error plays a critical role. Misincorporation of nucleotides during genome replication leads to the appearance of single nucleotide polymorphisms (SNPs) and insertions/deletions (InDels) both *in vivo* and *in vitro*. Previous studies proposed a very low mutation rate for HSV-1 polymerase ($1 \times 10^{-7}$ to $1 \times 10^{-8}$ mutations per base per infectious cycle), despite being performed on a single gene analysis located at a unique coding region [11,24,25]. However, that mutation rate does not correlate with the rapid ability observed when selecting HSV variants under drug pressures, either by selection of preexisting minor variants (MVs) in the population or by *de novo* mutations [24,26–29].

Herpesviruses were presumed to generate lower genetic diversity when replicating and then evolve with a slower rate than RNA viruses, due to their relatively more stable DNA genome and proofreading mechanisms [12,30,31]. However, over the last decade, the idea that herpesviruses exist as heterogenic and dynamic populations *in vivo* has gained increasing supportive evidence, aided by worldwide accessibility and cost reduction of high-throughput sequencing technologies [32–35]. Nonetheless, how this increasingly evident genetic variability is generated remains not well characterized. Studies conducted to understand the high diversity of human cytomegalovirus intrahost infections, concluded that this diversity was similar in mixed and single infections [36–38], while others have argued that it would be caused by coinfection with multiple distinct strains [35,39,40]. Shipley et al. reported that the genetic diversity of HSV-1 is able to change over multiple cycles of latency and reactivation in the genital context [41]. Clinical isolates of HSV-1 in Finland [32], and of HSV-2 from a neonatal population [33], showed extensive intra-host diversity, displaying different *in vitro* phenotypes among isolates. A similar level of generation of genetic diversity may happen *in vitro* as well, where the method of preparation/propagation of HSV stocks plays a key role [42]. In this context, multiple rounds of amplification in cell culture increase the heterogenicity of viral stocks by genetic drift, where plaque-isolation has been classically used to reduce the impact of this evolutionary force. HSV has been routinely propagated in African green monkey kidney (Vero) cells, which are interferon incompetent [43]. This, together with the fact that these cells are not of human origin and just represent only one cell type among all the ones that HSV faces when infecting humans, very likely impact the generation of genetic diversity *in vitro*. On the other hand, *in vitro* studies of viral evolution can bring an ideal scenario with more controlled and reduced selective pressures, where gaining insights into the mechanisms governing the generation of genetic variability can be easier. Kuny et al. [44] have recently reported that a heterogeneous

population of HSV-1 increased its genetic diversity and changed its phenotype more dramatically than a plaque-purified population of the same strain, after ten passages in Vero cells. Certain genetic MVs already present in the heterogeneous population appeared to be positively selected, whereas the purified population did not appear to increase its MVs repertoire. However, the number and frequency of the *de novo* mutations that appeared over those ten sequential passages in Vero cells were not assessed. As far as we know, similar studies reporting how HSV-2 *in vitro* evolves in terms of generation of genetic diversity have not been reported yet. Thus, how differentially fast this genetic diversity is *de novo* generated *in vitro*, and how it impacts the potential coding diversity (and thus, evolution) between HSV-1 and HSV-2, remains unclear.

Here we performed sequential passage of two plaque-purified populations of both HSV-1 and HSV-2 to characterize the differential ability and speed of these closely related viruses to *in vitro* evolve in two different cell lines. We first assessed the genetic diversity present in our original viral stocks by high-throughput sequencing, isolating plaque-purified clones, and sequencing them. After nonconservative variant analysis, two purified clones were ultra-deep re-sequenced, characterized in cell culture, and tested in animal models of infection; in order to ensure that the genetic bottleneck of the plaque-purification procedure did not alter the viral infectiousness due to a deleterious variant unintentionally selected [45–47]. After establishing a high confident baseline of the genetic diversity present in each purified population at passage 0 (P0), each of them was subjected to ten serial passages in Vero cells and human keratinocytes (HaCaT cells), and ultra-deep sequenced at passage 5 (P5) and passage 10 (P10). We detected the frequency and distribution of preexisting and *de novo* genetic MVs, examining their impact on the coding capacity of both HSV serotypes. These results have helped us to better understand how each HSV subtype differentially evolves, depending on the selective pressures behind a given cellular environment, and brings new insights into the different generation of genetic diversity of the closely related HSV-1 and HSV-2.

## Results

### The viral phenotype of plaque-purified clones may dramatically change just in ten passages in cell culture

To better understand the generation of genetic diversity in HSV-1 and HSV-2, we first isolated five viral clones from both HSV-1 strain SC16 and HSV-2 strain 333 original stocks after five rounds of sequential plaque purification in Vero cells (Fig 1A). This was done in order to reduce the preexisting genetic variability present in the original viral populations, which was used as the baseline to assess the generation of genetic variability.

Both original stock and purified clones, for each serotype, were deep-sequenced (S1–S3 Figs, see Sheet A in S1 Table for genome sequence statistics). After variant analysis, two purified clones for each subtype were selected: clones 2 and 3 for HSV-1, and clones 1 and 5 for HSV-2. This selection was based on the lowest degree of nonconservative variability compared to their corresponding reference sequence, previously described as *de novo* assembled consensus genomes of the same original stocks used in this study [48,49]. We compared the virus growth kinetics in cell culture, as well as the infectivity and pathogenesis of these selected purified clones to their parental populations in mouse models of infection, in order to ensure that the MVs detected in the purified populations did not affect its viral fitness significantly (S4 Fig and S1 Text). These selected purified clones were then ultra-deep re-sequenced (referred to as P0, Fig 1A), allowing us to accurately identify very low frequency MVs present in these purified populations (S5 Fig). That was a critical step to establish a high confidence baseline to be

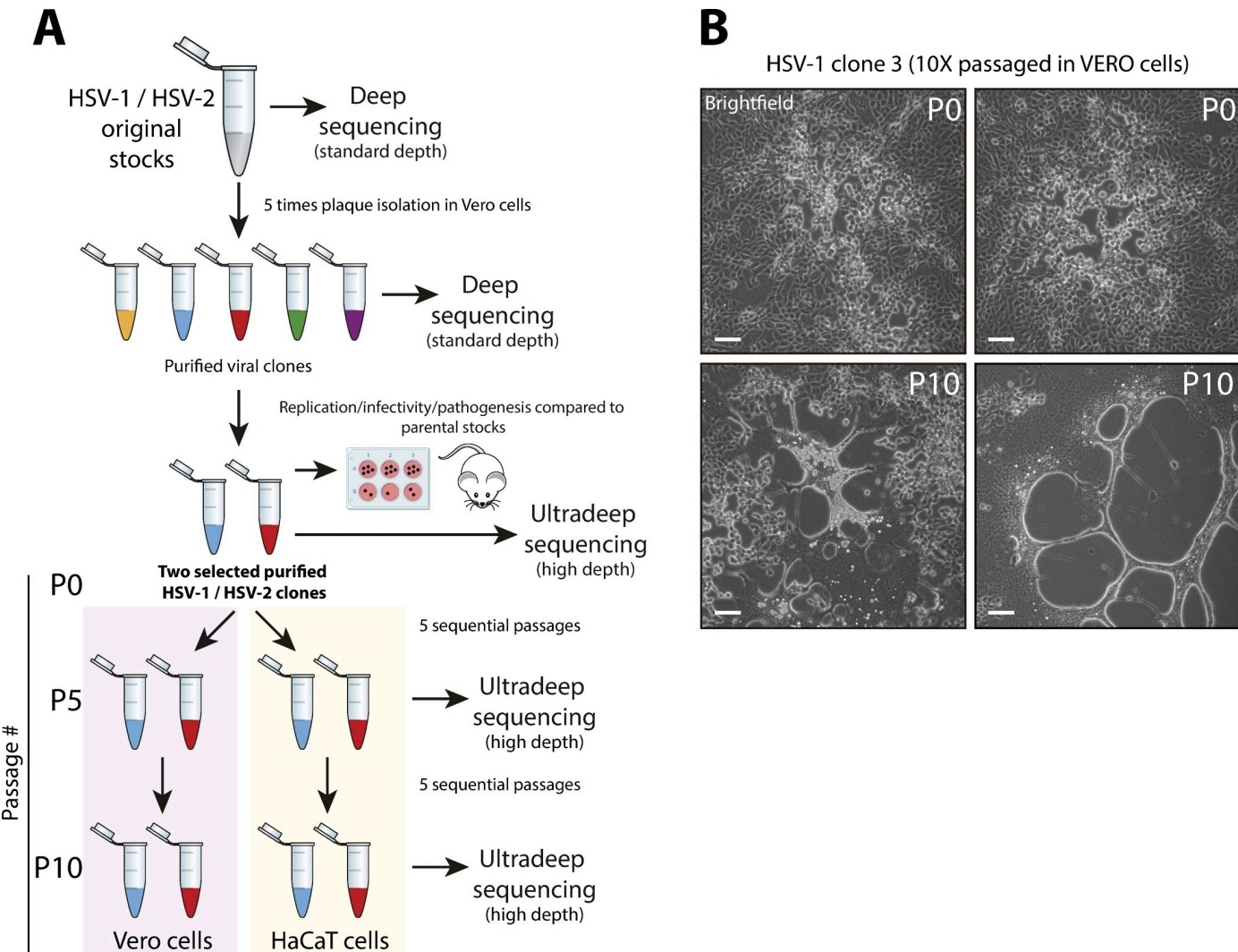

**Fig 1. Experimental design of the *in vitro* generation of genetic variability studies for HSV-1 and HSV-2 subtypes and detected changes in plaque phenotype.**
(A) Five viral clones from each original stock were five times plaque-purified in Vero cells and then deep sequenced. Two clones were re-sequenced at ultrahigh depth, whose replication, infectivity, and pathogenesis were compared to their corresponding parental stocks in cell culture and animal models of infection (S4 Fig). Those two clones were used to infect Vero and HaCaT cells at an MOI of 0.1 PFU/cell. After 48 hpi, viral progenies were harvested, referring to this infection cycle as a passage. Viral populations from each plaque-purified clone were ultra-deep sequenced after five and ten passages in each cell line. (B) Four representative pictures are showing the plaque morphology phenotype of HSV-1 clone 3 in Vero cells (48 hpi), before (passage 0, P0), and after ten passages (P10) in Vero cells. This syncytial plaque phenotype is due to the previously well-described syncytia-inducing mutation in UL27 CDS (see Sheet I in S1 Table, variant #38: R858H) [44]. Tiled images (4 x 4) were taken using a Leica DM IL LED inverted microscope equipped with a Leica DFC3000-G digital camera. Scale bars indicate 100 μm.

able to discriminate between preexisting or *de novo* generated MVs, due to the intrinsic variability generation rate in each HSV serotype replication cycle.

Each purified clone was used to infect a separate monolayer of Vero and HaCaT cells at a multiplicity of infection (MOI) of 0.1 PFU/cell. Once cytopathic effect was observed, cells and supernatant were harvested. This cycle of infection is considered as a passage in the generation of variability experiments. The viral stock from the first passages of each selected purified clone was used to infect the next monolayer of Vero and HaCaT cells at the same estimated MOI, being this process repeated for ten sequential passages (Fig 1A, lower part). The viral population of each purified clone in each cell line was ultra-deep sequenced after five and

finally ten sequential passages (S6–S9 Figs). An MOI of 0.1 PFU/cell was selected to allow for multiple rounds of replication in each passage, favoring the generation of genetic variability in the viral population.

Kuny et al. [44] did not find any changes in plaque phenotype in an HSV-1 purified population passaged ten times in Vero cells. However, we observed dramatic changes in plaque morphology in the purified HSV-1 clone 3 after ten passages in Vero cells (Fig 1B). After ultra-deep sequencing and variant analysis, we found a nonsynonymous SNP previously described as the cause of the syncytial plaque phenotype [50–52]. The frequency of the R858H variant in the UL27 gene (encoding glycoprotein B, gB) was non-existent at P0 and P5, reaching 46.52% at P10, just in 5 passages in Vero cells (Sheet I in S1 Table, variant #38). In addition, the purified HSV-2 clone 1 also exhibited the same variant at P5 in Vero cells, with a frequency of 1.75% and being undetected at P0 (Sheet K in S1 Table, variant #64). In contrast with HSV-1 clone 3, the frequency of this variant did not increase at P10 for HSV-2 clone 1, not acquiring the syncytial plaque phenotype. Neither this variant nor other syncytia-inducing MVs in gB, such as L817P [44], were detected when purified clones were passaged in HaCaT cells, for both HSV subtypes. These results suggested that even purified viral populations with low diversity are able to quickly change or evolve just in a few passages in cell culture, being more prone to happen in some cell lines than in others.

## The high genetic diversity found in HSV-1 and HSV-2 original stocks was significantly reduced in five rounds of plaque isolation in Vero cells

Genetic diversity can be defined as nucleotide alleles or variants present in a given percent of the sequencing reads, at a given locus in a sequenced viral population. With enough deep sequencing coverage, these MVs can be confidently detected, revealing the genetic diversity present in the viral population. HSV-1 and HSV-2 original stocks had an average coverage depth of 1783 and 1320 reads/position, respectively (Sheet A in S1 Table). We identified SNPs and InDels that were present in greater than 1 percent of the sequencing reads (1 percent cut-off as the threshold of detection, plus additional coverage-dependent filters, see "Material and methods" for detailed criteria). Both viral populations from original stocks had MVs at different sites and frequencies, being evenly distributed across highly repetitive areas and coding regions into their reference genome (S1 Fig). As previously described for other mixed populations of HSV [33,44], both HSV-1 and HSV-2 original stocks displayed a significant number of MVs (Fig 2A), when sequencing reads were aligned to each corresponding reference genome (*de novo* assembled consensus genome of a purified clone from each same parental stock [48,49]). HSV-1 original stock registered a total of 712 MVs, whereas HSV-2 stock, 1044 (Fig 2A, see Sheets C and D in S1 Table for full lists of MVs). Detailed analysis after variant calling showed that the higher fraction of MVs corresponded to nonsynonymous SNPs for both HSV-1 and HSV-2, accounting for 452 and 701, respectively (Sheet B in S1 Table). In terms of frequency, the major fraction of total detected MVs showed to be between 1% - 10% for both HSV serotypes (Fig 2B).

Because we found a high level of genetic variability in sequenced viral population from each original stock, we plaque-purified five viral clones from each parental stock in order to use low genetically diverse viral populations as a baseline to evaluate the generation of *de novo* variability in both HSV subtypes. After sequential plaque-isolation of independent viral clones from their parental stocks, sequencing data from these purified viral populations revealed a significant reduction in the number of total detected MVs, for both HSV-1 and HSV-2 (Fig 2C, see Sheet C and D in S1 Table for full lists of MVs). In fact, every purified clone showed a significant reduction in the total number of MVs, proportionally reflected in the number of

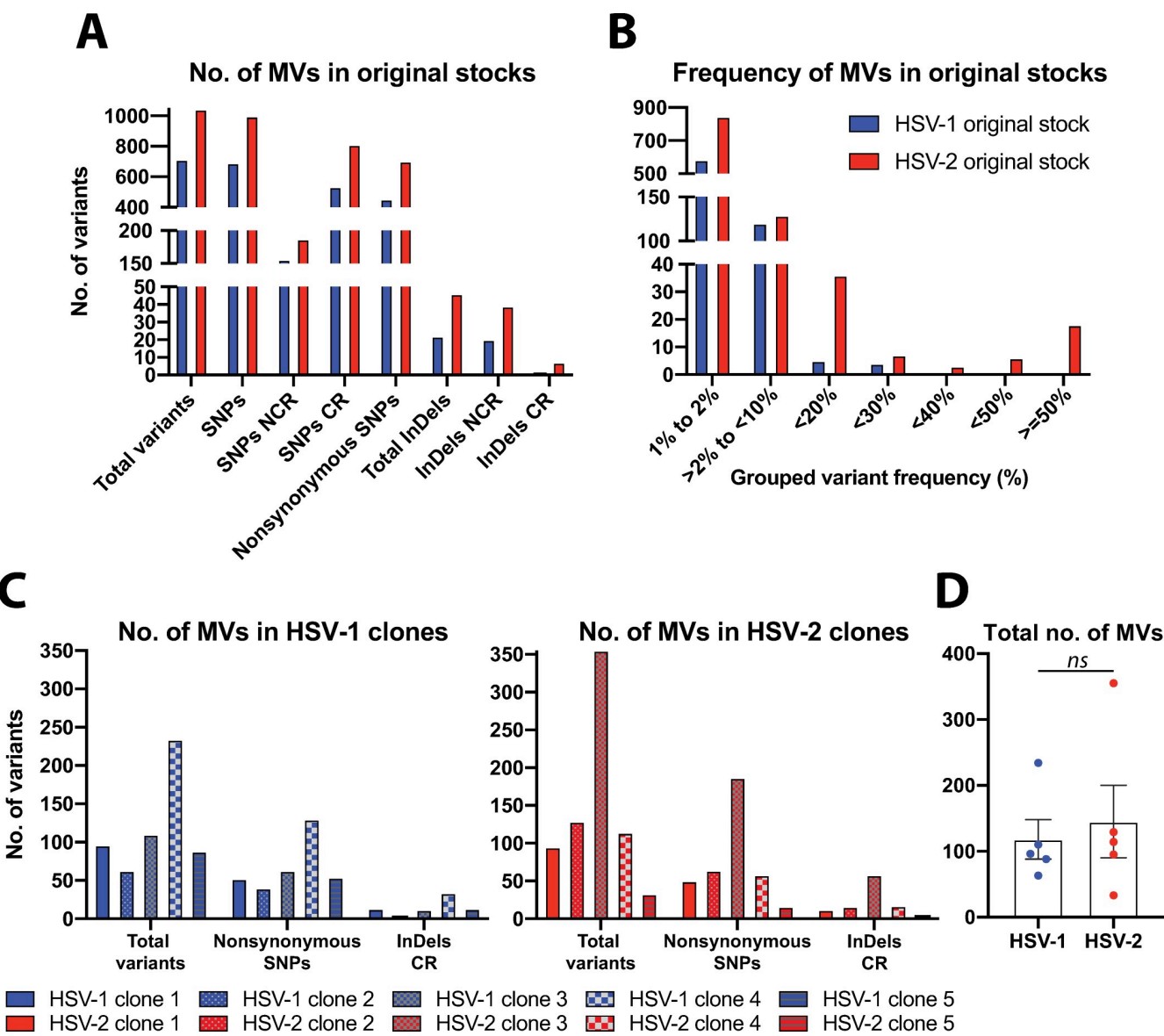

**Fig 2. Genetic diversity in viral populations from original stocks and five times plaque-isolated HSV-1 and HSV-2 clones.** (A) Total number of MVs observed in each original stock, at a frequency equals or above a 1% limit of detection (see Material and Methods for details). The total number of MVs (*y*-axis) is separated into variant type and genomic location (*x*-axis). Variant type distinguishes between SNPs and InDels, discriminating nonsynonymous SNPs. The genomic location of each variant is categorized as non-coding or coding region. (B) Histograms show the number of MVs in each frequency range for each original stock. The frequency of each variant was examined and grouped in shown ranges (e.g., 10% to <20% frequency, 20% to <30% frequency and so on). SNPs and InDels were combined for this analysis. (C) Total number of MVs observed in each HSV-1 (left, blue bars) and HSV-2 (right, red bars) clones after five rounds of plaque-isolation, categorized by nonconservative changes (SNPs and InDels). See Sheets C and D in S1 Table for full lists of MVs position and frequency data. (D) Total number of variants detected in each clone are grouped by subtype and graphed showing mean +/- SEM (ns = not significant *p* > 0.05 by two-tailed Mann–Whitney *U*-test).

nonsynonymous SNPs, for both serotypes. With the exception of HSV-1 clone 4, as well as HSV-2 clone 3, every other purified clone reduced its genetic variability by 10-fold or higher, when the total detected number of MVs was compared to the number registered for their original stocks. The total number of MVs detected from each clone was grouped by HSV subtype and compared between them, showing no statistically significant differences (Fig 2D), despite the differences in average coverage depth among them (Sheet A in S1 Table). Based on the

lowest number of nonconservative changes (i.e., nonsynonymous SNPs and InDels in coding regions) observed among the five purified clones for each HSV subtype, as well as their frequencies in the viral populations, two purified clones of each serotype were selected for further characterization and *in vitro* evolution studies. HSV-1 clones 2 and 3 were selected and tracked by a differential SNP in the UL14 CDS (#162, Sheet C in S1 Table), while HSV-2 clones 1 and 5, by a differential SNP in the UL13 CDS (#180, Sheet D in S1 Table).

The selected purified clones were used to perform replication kinetics in Vero cells, as well as to infect mice as described in S1 Text, in order to confirm whether the MVs detected could cause a deleterious effect in terms of infectivity and pathogenesis, compared to their parental stocks (S4 Fig). No significant differences were found in the replication and infectivity of the purified clones (S4A Fig). Despite finding some variability in terms of survival when compared to original stocks, all four selected clones were able to successfully infect and cause disease in both mouse models of infection tested (S4B Fig), understanding that none of the unintentionally selected MVs during the plaque-purification genetic bottleneck caused a significant deleterious effect. Thus, plaque-isolation proved to be a successful approach to decrease the genetic diversity in the purified viral populations before studying the generation of genetic variability in cell culture. These purified viral populations of more uniformed genetic diversity constituted the key starting point to determine with high confidence the generation of genetic variability, particularly in terms of very low frequency MVs.

## Depth of sequencing is critical to establish a high confidence baseline in order to detect very low frequency genetic diversity

Depth of sequencing coverage is instrumental in detecting MVs with high reliance and accuracy. Because the number of sequencing reads correlates directly with the frequency of alleles in the viral population, a higher depth of coverage allows a better resolution of the genetic diversity present in a given viral population. Notably, when identifying *de novo* appeared MVs, a high depth of coverage is crucial to detect the genetic diversity represented with a very low frequency in the viral population (i.e., theoretically at least an average coverage of 200 reads/ position, a hundred paired-end reads, would be required in order to be able to detect a 1 percent frequency variant, supported by two reads contained the same allele).

Nonetheless, having an average coverage depth of 200X from a sequenced viral stock containing $10^6$ PFU, it would only represent 0.02 percent of the viral population. To surmount the fact that the sequencing of a minute fraction of the whole viral population might seriously bias our ability to detect a given variant, and therefore to determine its novelty, we ultra-deep resequenced the previously selected purified viral clones. The average coverage depth increased approx. 2 logs for each purified viral clone, from $10^2$ (standard-deep sequencing, SDS) to $10^4$ (ultra-deep sequencing, UDS) (Fig 3A and Sheet A in S1 Table). In this context, this depth of coverage would represent a theoretical 1 percent of the viral population (assuming $10^6$ PFU), having increased a hundred times the actual genetic diversity sampled from each viral population. Based on this, we understood this level of coverage depth constituted a high confidence representation of the genetic diversity present in each viral population.

A more detailed look at the sequencing statistics of the ten purified clones aforementioned showed that among each serotype, HSV-1 clone 4 and HSV-2 clone 3 showed the highest depth of average coverage, as well as the highest number of total MVs (Fig 2C and Sheet A in S1 Table). This could reasonably lead to think that the lower number of detected MVs in the selected purified clones was due to their lower average coverage. Nevertheless, we did not find a dramatic increment in the total number of MVs obtained from SDS versus UDS for any of the four selected purified viral clones (Fig 3B and Sheet B in S1 Table). The total number of

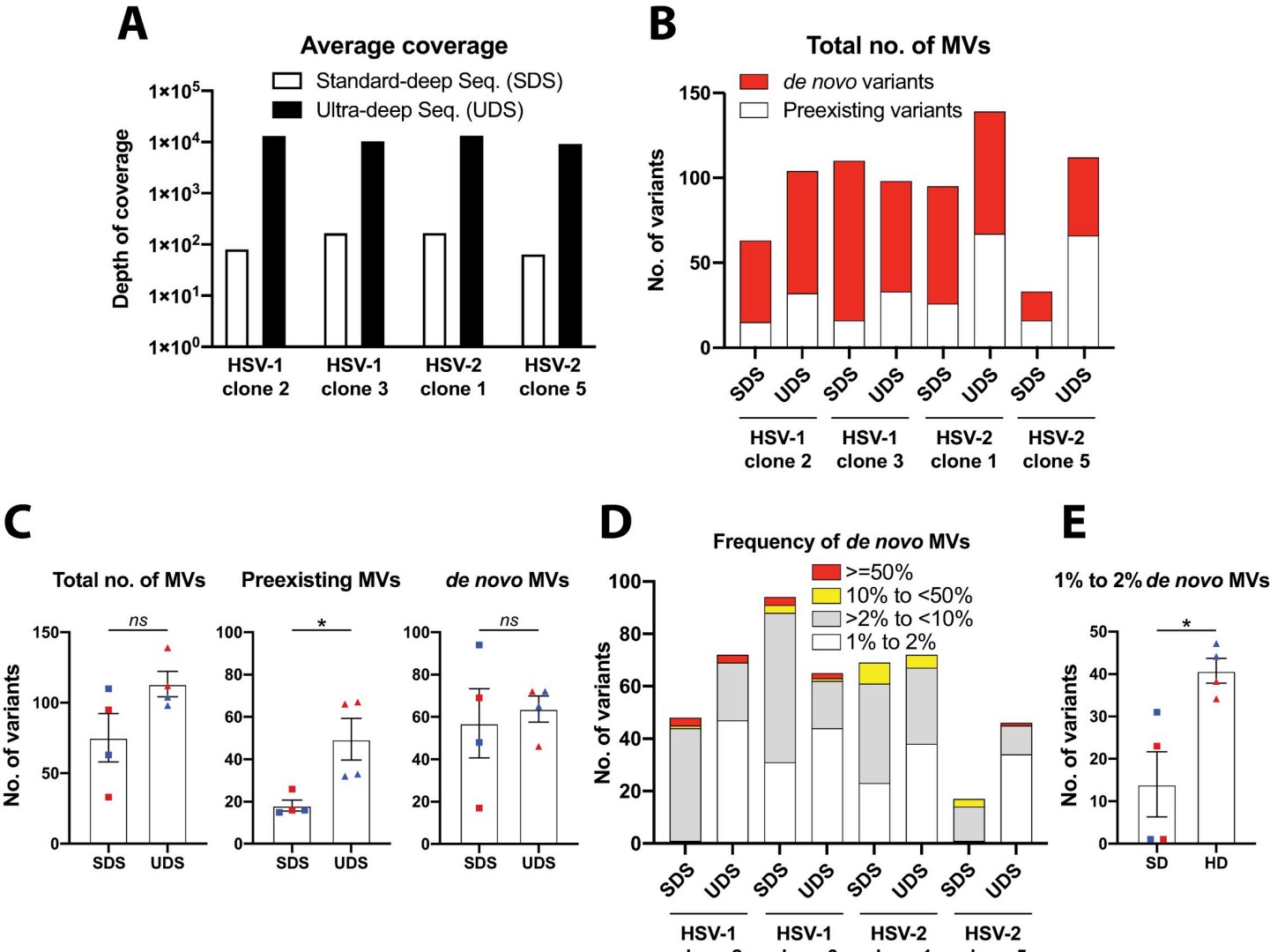

**Fig 3. Comparative variant analysis from standard- and ultra-deep sequencing data of two plaque-isolated HSV-1 and HSV-2 clones.** (A) Histograms show the average depth of coverage per genomic position of reads alignments from standard-deep sequencing (SDS, white bars) and ultra-deep sequencing (UDS, black bars) for each viral clone. (B) Histograms bar plot total number of MVs observed after variant analysis of SDS and UDS data in each viral clone, discriminating between preexisting MVs found in the corresponding original stock (white) and *de novo* appearance (red). (C) The number of total, preexisting, and *de novo* MVs detected in each clone from SDS, and UDS data are grouped and graphed (blue shapes for HSV-1, red for HSV-2 clones), showing mean +/- SEM (ns $p > 0.05$, * $p = 0.029$ by two-tailed Mann–Whitney $U$-test). (D) Number of *de novo* MVs observed after variant analysis of SDS and UDS data in each viral clone, stacked by frequency ranges. SNPs and InDels were combined for this analysis. (E) *de novo* MVs with a frequency between 1% to 2% detected in each clone from SDS, and UDS data are grouped and graphed (blue shapes for HSV-1, red for HSV-2 clones), showing mean +/- SEM (* $p = 0.029$ by two-tailed Mann–Whitney $U$-test). See Sheets E and F in S1 Table for full lists of MVs position and frequency data.

MVs from SDS versus UDS were grouped and compared, showing no statistically significant differences (Fig 3C left graph), while grouped preexisting MVs did (Fig 3C center graph). The number of preexisting MVs (i.e., variants already detected in the original parental stock) increased in HSV-1 purified clones, being this increment higher in HSV-2 clones (Fig 3B). However, when *de novo* MVs were grouped, there was no statistically significant difference between them (Fig 3C right graph), despite all but HSV-1 clone 3 showing an increment in the total accounted *de novo* MVs (Fig 3B and Sheet B in S1 Table). When we grouped these *de novo* MVs based on their frequency, from SDS versus UDS data, we observed a consistent

increase in the number of very low frequency MVs (1% to 2%), as well as a reduction in the number of low frequency MVs (>2% to <10%), across every purified clone (Fig 3D and Sheet B in S1 Table). This reduction in the number of *de novo* low frequency MVs was particularly pronounced in the case of HSV-1 clone 3, explaining the observed decrease in the total number of *de novo* MVs foresaid. Because of the higher depth of average coverage from UDS, we were able to detect a significantly higher number of *de novo* MVs with a very low frequency in the viral populations (Fig 3E), but also to better discriminate the actual frequency of these MVs in the viral population. These improvements of UDS may explain the reduction in the number of low frequency MVs, switching to be detected with a lower frequency when the deep sequencing coverage increases. The fact that the number of preexisting MVs, but not the number of de novo MVs, increased dramatically between SDS vs. UDS indicated that higher depth of coverage helps to gain accuracy and resolution characterizing the existing genetic diversity. Since all four purified clones showed a comparable level of genetic diversity, in terms of total and *de novo* MVs (ranging from 98 to 139, Sheet B in S1 Table), we determined this as a suitable, high confidence starting point to study the generation of genetic variability of both HSV subtypes in cell culture.

## HSV-2 *in vitro* evolves dramatically faster than HSV-1 in both Vero and HaCaTs cells

Having characterized the genetic diversity present in each purified viral clone with high confidence, we next investigated how HSV-1 and HSV-2 differentially evolve in cell culture. We conducted ten sequential passages of each purified viral clone separately in Vero and HaCaT cells, ultra-deep sequencing each viral population after 5 and 10 passages in each cell line. We obtained a broad range of average coverage depth, ranging from 872 to 25892 reads/position (Sheet A in S1 Table). Nonhuman primate kidney-derived epithelial (Vero) cells are widely and routinely used for HSV propagation, whereas human keratinocyte (HaCaT) cells are closer to the natural physiology of HSV infection in the skin. HSV plaque formation, cell-to-cell spread, and cell migration were reported to be significantly different when compared HaCaT to Vero cell infections [53]. Based on that, we sought to examine the effects of the differential selective pressures present in each cultured cell line to the generation of genetic diversity of each HSV subtype.

After variant analysis, we analyzed the total number and types of MVs that were present in each viral population at P5 and P10, for every purified clone passaged in each cell type. Most of the detected MVs corresponded to mutations impacting coding regions, including SNPs and InDels (see Sheet B in S1 Table). Nonetheless, both HSV-1 and HSV-2 populations displayed a similar fraction of the total number of MVs impacting coding regions, around 56 percent (+/- 0.094% SD) on average (i.e., nonsynonymous SNPs plus InDels in coding regions, divided by the total number of detected MVs, see Sheet B in S1 Table). We detected a consistently higher total number of MVs among HSV-2 than in HSV-1 viral populations, as well as a higher and increasing number of *de novo* MVs after sequential passages, not previously detected in the purified viral populations at P0 (Fig 4A and S1–S4 Animations). Preexisting variability in HSV-1 populations remained constant over the ten passages in both cell types, but neither a relevant increase in the total number of MVs nor in *de novo* generated MVs were observed (Fig 4A left graph, see Sheet B in S1 Table for more details). However, despite remaining the preexisting number of MVs constant in HSV-2 populations over sequential passages, we found a consistent increment in the appearance of *de novo* generated MVs after five, and even greater, after ten passages for every purified clone (Fig 4A right graph). Both HSV-2 purified clones showed the most drastic increment in the number of *de novo* MVs after being

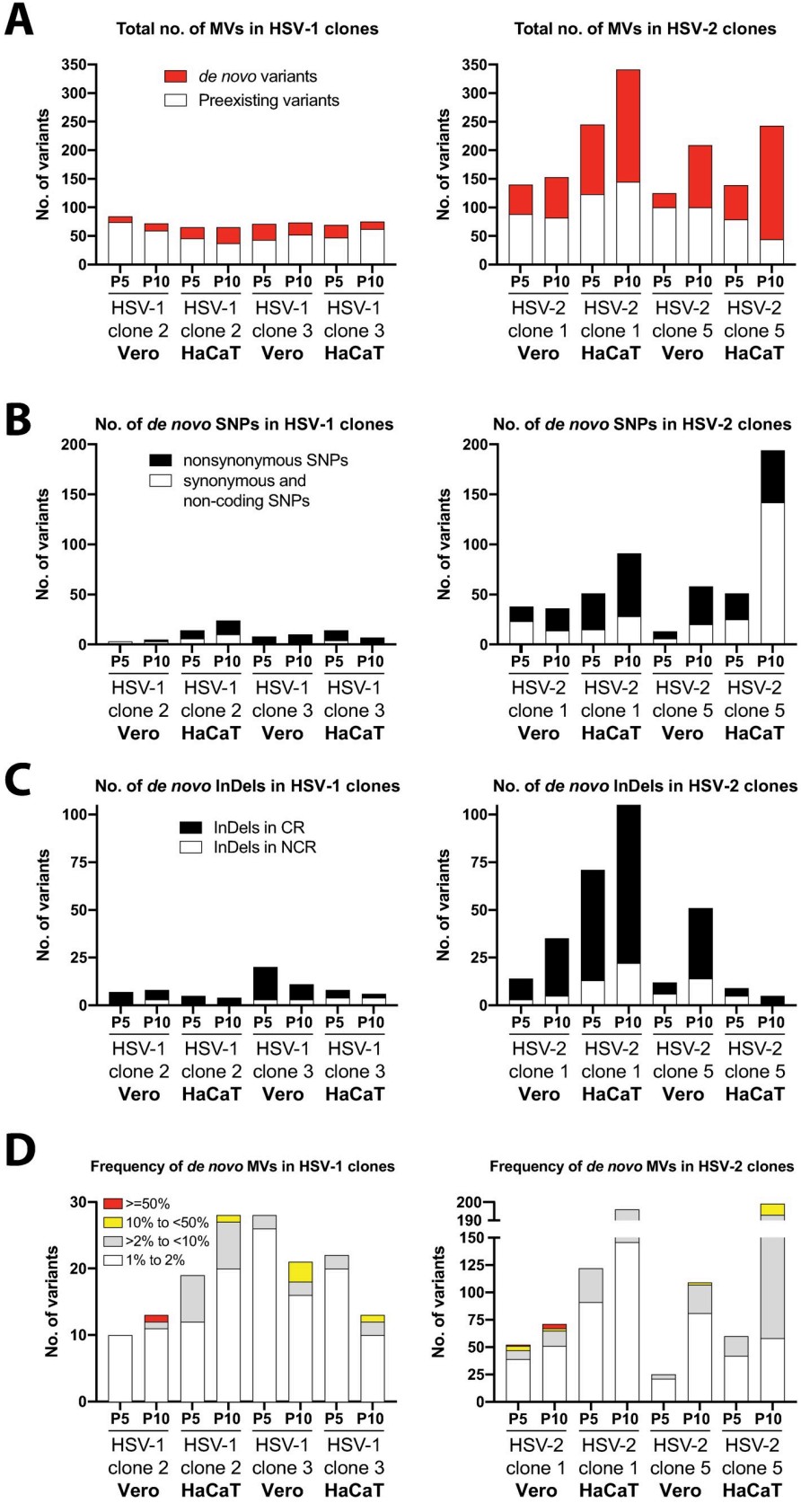

**Fig 4. Comparison of *de novo* generation of total and nonconservative genetic diversity between HSV-1 and HSV-2 purified clones, after five (P5) and ten (P10) passages in Vero and HaCaT cells.** (A) Total number of MVs are plotted according to variant analysis data for each clone, passage, and cell line, differentiating between preexisting MVs observed in the corresponding passage zero (P0) in white, and *de novo* generated MVs in red. (B) Number of *de novo* SNPs detected in each viral population, categorized by nonsynonymous (black) and synonymous/non-coding changes (white). (C) Number of *de novo* InDels detected among each viral population are stacked by their location impacting coding regions (black) or non-coding regions (white). (D) *De novo* MVs are stacked by frequency ranges. SNPs and InDels were combined for this analysis. See Sheets G-N in S1 Table for full lists of MVs position and frequency data.

subjected to ten sequential passages in HaCaT cells, dramatically higher than that found when sequentially passaged in Vero cells (Fig 4A right graph). Additional analyses using different software packages for variant calling showed similar results, despite the reduction in total number of MVs detected between methods (Sheet O in S1 Table); where HSV-2 clones consistently exhibited greater rates of MVs accumulation than HSV-1 counterparts across software approaches (S10 Fig and Sheet P in S1 Table). These results robustly showed that HSV-2 generates *de novo* genetic diversity faster than HSV-1, where the selective pressures present in each cell type used for viral propagation may differentially affect how the genetic diversity is generated. In a more detailed analysis of these *de novo* MVs, we classified them as substitutions or SNPs and InDels. We found that the *de novo* MVs generated over sequential passage of both HSV-1 and HSV-2 purified populations were predominately nonconservative changes (Fig 4B and 4C). That was particularly remarkable in the case of HSV-2 populations, where the number of detected nonsynonymous *de novo* SNPs and InDels impacting coding regions gradually and consistently increased, over the ten passages in both cell types and for both purified clones (Fig 4B and 4C, right graphs). These data suggest that the generation of *de novo* genetic diversity in HSV promotes predominantly nonconservative changes.

For each *de novo* variant, we also examined its frequency in the population. They were clustered, based on their frequency, in very low frequency MVs (1% to 2%), low frequency MVs (>2% to <10%), medium frequency MVs (10% to <50%), and high frequency variants (equal to or greater than 50%). We observed that the major fraction of *de novo* generated variability corresponded to very low frequency MVs, consistently displayed by almost every purified clone after ten passages in both cell lines (Fig 4D). These very low frequency MVs barely increased in HSV-1 populations after sequential passages, even slightly decreasing as displayed by HSV-1 clone 3, whereas HSV-2 clones showed a consistent and significant increment of those. As each HSV-2 population was passaged, the very low frequency MVs increased their proportion in the population as the predominant group of MVs. Additionally, we also observed that there was a gradual and systematic increment of low-medium frequency MVs (taken together) across every viral population of both HSV serotypes (Fig 4D). In this regard, we found that both HSV-2 purified clones remarkably showed the highest increment of low-medium frequency MVs when passaged in HaCaT cells, where HSV-2 clone 5 increased the proportion of these low frequency MVs over the total number of *de novo* variants, from 30% (at P5) to 67% (at P10) (Fig 4D right graph, see Sheet B in S1 Table for details). These data identify these very low frequency MVs as the main source of generation of genetic variability in HSV, gradually increasing as a percentage of the viral population over sequential passage, where each HSV subtype changes with a different speed in response to the same pressures of a given environment.

### *De novo* variants increase the potential coding diversity of HSV-2

After observing that the *de novo* genetic diversity detected after sequential passages in culture was predominantly translated into nonconservative changes for both HSV subtypes, we

further examined the distribution of *de novo* MVs impacting coding regions. In accordance with the total number of *de novo* MVs aforementioned (Fig 4A), HSV-1 viral populations displayed an anecdotic low number of MVs impacting just a few genes, while nearly every HSV-2 gene harbored *de novo* MVs (Fig 5). Despite finding an even distribution of *de novo* MVs across the coding genome, some HSV-2 genes were identified as hotspots of novel coding variability (e.g., UL16, UL27, UL28, UL29, UL36). It is also remarkable to observe how the sequential passage of both HSV-2 purified clones in HaCaT cell impacted much more heavily the coding capacity of the HSV-2 genome, as shown in Fig 5, where yellow and orange colors (purified clones passaged in HaCaT cells) dominate over blue tones (passages in Vero cells). Although RL1, RL2, and RS1 genes are located into repetitive areas of the genome, which have been demonstrated to be regions of high variability [45,47], we also identified a higher number of *de novo* MVs present in these genes after sequential passage of HSV-2 clones in HaCaT cells, compared to Vero cells (Fig 5). As shown in Fig 4B and 4C, for both *de novo* appeared SNPs and InDels, intragenic MVs outnumbered those in intergenic regions, despite thinking that higher selective pressures would reduce the appearance of unfavorable mutations in coding regions. These data revealed the impact of very low (1%-2%) and low frequency (<10%) *de novo* MVs in expanding the coding genetic variability in HSV-2, which might have differentially contributed to HSV-2 evolution depending on specific niches.

## Frequency increase of *de novo* variants may depend on positive selective pressure in cell culture

Finally, we examined how the frequency of nonconservative *de novo* variants changed over sequential passages of each purified viral population in both cell types. We found that most of these nonconservative *de novo* MVs did not increase their frequency in the population, seeming to reach a stationary equilibrium or just disappearing after ten passages (Sheets G-N in S1 Table). We did not detect any *de novo* InDel in coding regions gradually increasing its frequency in the passaged viral populations, likely reflecting the stronger selective pressure against missense mutations in coding regions. However, a few nonsynonymous *de novo* SNPs were found increasing their frequency significantly in the viral populations, which suggested that those might be conferring a selective advantage. The clearest example for this outcome is illustrated by the syncytia-forming MVs in UL27, as previously described by others [44,50–52], also observed in the purified HSV-1 clone 3 passaged in Vero cells (Fig 1B). That variant in UL27 (R858H) was not high-confidently detected neither at P0 nor at P5 in Vero cells, *de novo* appearing after P5 and reaching almost a frequency of 50% in the viral population at P10 (Fig 6, left graph, and Sheet I in S1 Table). On the other hand, we also observed an interesting positive selection example of a *de novo* variant not conferring a selective advantage in cell culture. We found a nonsynonymous *de novo* variant in the UL13 gene with a frequency of 67% within the HSV-2 clone 1 population after five passages in Vero cells, reaching 90% of the population at P10 (Fig 6, right graph, and Sheet K in S1 Table). It has been described that UL13 kinase activity is required for axonal transport *in vivo* [54], but it is dispensable *in vitro* [55]. Missense mutations in the UL13 gene have been reported to increase in frequency in different strains of HSV-1 [21,23,44,45]. Thus, it is reasonable to think that the UL13 variant found in HSV-2 clone 1 was then not selected based on a selective advantage, but because of the previously observed high tolerance of UL13 inactive kinases. Other MVs, such as those impacting UL14 and UL24 in HSV-2 populations (Fig 6), also reached almost 80–90% of the population at P10. Different variants impacting those genes in HSV-1 have been described, increasing their frequency over sequential passage in cell culture [44], which suggests that nonconservative

## HSV-1 genes impacted by *de novo* variants

## HSV-2 genes impacted by *de novo* variants

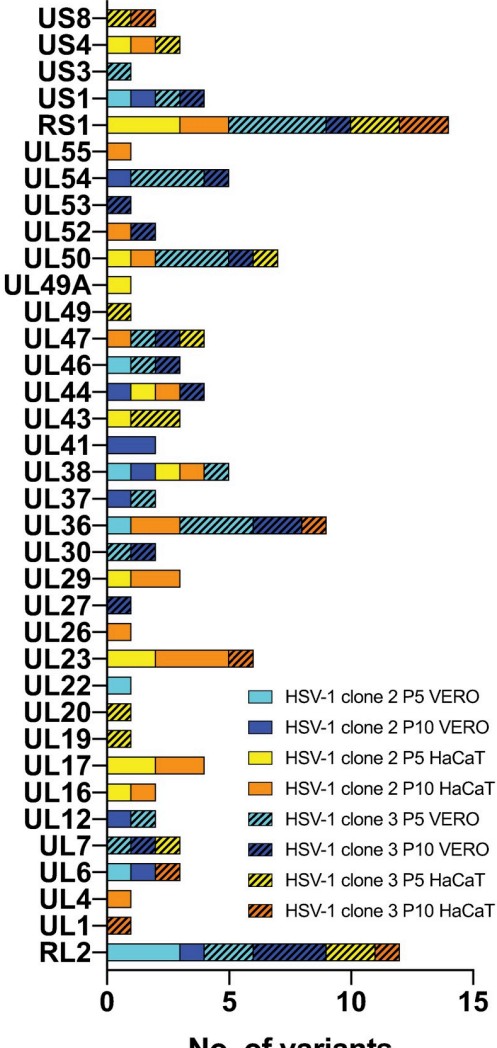

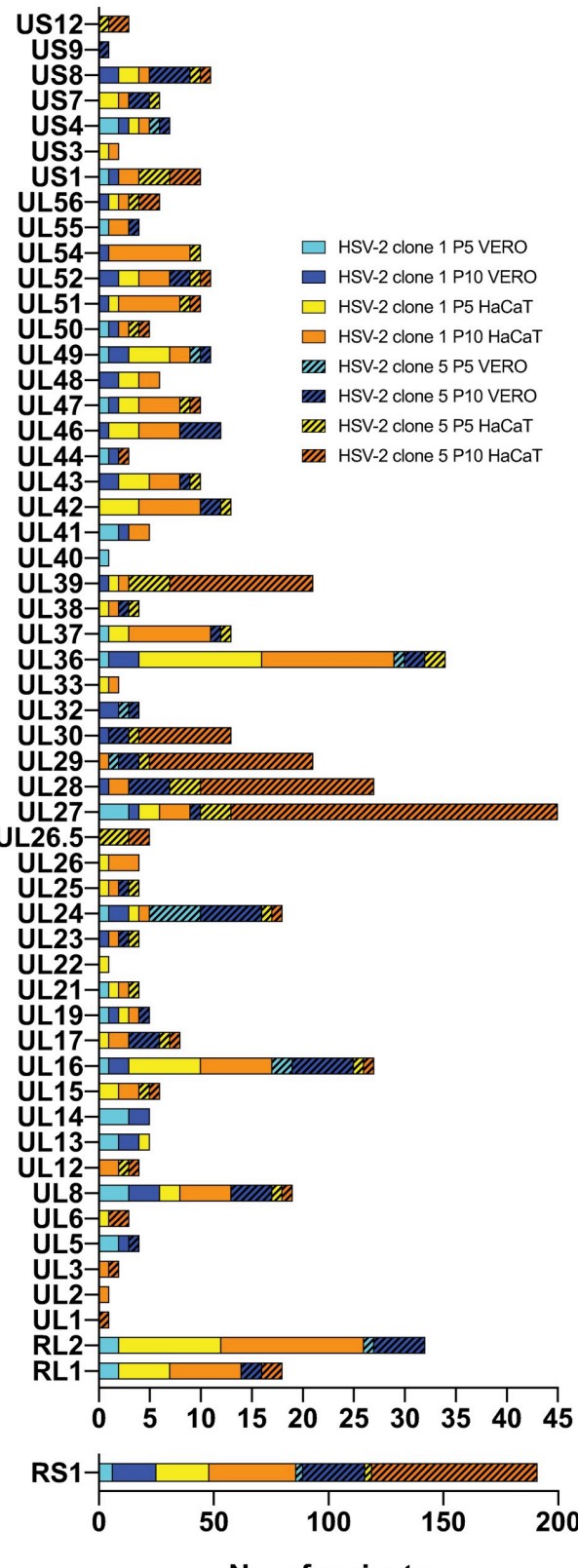

**Fig 5. Stacked histograms show the number of *de novo* genic MVs (*x*-axis) located in each HSV-1 (left) and HSV-2 (right) coding sequence (gene; *y*-axis), after five (P5) and ten (P10) passages in Vero and HaCaT cells.** Only coding sequences registering at least one variant are included in the histogram. MVs found in both copies of each RL1, RL2, and RS1 coding sequences are listed together. SNPs and InDels were combined for this analysis. See Sheets G-N in S1 Table for full lists of MVs position and frequency data.

MVs affecting those genes might confer a selective advantage, or just be more tolerable when HSV replicates *in vitro*.

## Discussion

In this study, we assessed the *de novo* evolution of HSV-1 and HSV-2 generated over sequential passage in two different cell types, using the neurovirulent viral strains SC16 and 333 as a model for each HSV subtype. We characterized for the first time the whole-genome generation of new genetic diversity during viral replication *in vitro*, after having set a high confidence baseline of the preexisting genetic diversity by ultra-deep sequencing of plaque-purified viral populations. This approach allowed us to identify very low frequency *de novo* mutations within genetically homogeneous viral populations, and then examine how viral populations of both HSV serotypes drifted under equal cell culture conditions, in two different cell types. We found that both HSV-1 and HSV-2 increased the number and frequency of *de novo* MVs after five and ten passages, being most of those low frequency nonconservative mutations impacting coding regions. Interestingly, we observed that purified HSV-2 populations were much more prone to generate genetic diversity during passaging than HSV-1, despite displaying a similar number of total preexisting MVs before sequential passages. While the genetic diversity of HSV-1 clonal populations remained similarly stable after being passaged in both Vero and HaCaT cells, HSV-2 purified clones evolved significantly faster when passaged in HaCaT than in Vero cells. HaCaT cells are human skin keratinocytes, an epithelial cell type closer to the natural physiology of HSV infection in the skin. HSV plaque formation, cell-to-cell spread, and cell migration were reported to be significantly different when compared HaCaT to Vero cell infections [53]. In addition, HSV-1 clinical isolates circulating in Finland replicated to lower titers and produced fewer extracellular viral particles in Vero than in HaCaT cells [32]. Since HSV propagation in Vero cells has been the traditional method for viral stock production and *in vitro* studies in virology labs, the evolutionary dynamic shift shown here by HSV-2 in each cell type highlights how critical it is to understand and characterize viral evolution and

**Fig 6. Dynamics of nonconservative *de novo* variants in each HSV-1 (left) and HSV-2 purified populations (right), whose frequency increased over sequential passages in cell culture.** Nonsynonymous *de novo* SNPs were plotted by their frequency in the sequenced viral population after five (P5) and ten (P10) passages in Vero and HaCaT cells. SNPs, their encoded proteins (bold), as well as the change that they would cause in the translated protein (italic), are listed in the legend according to their frequency at P5 and P10.

adaptation in specific cell type cultures. It is instrumental to determine how specific cell-type-associated selective pressures affect our experimental understanding of viral evolution and population dynamics *in vitro*.

From these *in vitro* studies, we monitored how genetic drift happened faster in HSV-2 than in HSV-1, emphasizing the differential contribution to generate genetic diversity under equal *in vitro* controlled conditions. This lower genetic variability showed by HSV-1 seems to be strain-independent, as recently reported for HSV-1 strain F [44]. Both heterogeneous and purified populations of this HSV-1 strain were sequentially passaged ten times in Vero cells. We quantified the number of *de novo* mutations that appeared over five and ten sequential passages using their available data, finding 19 (P5) and 7 (P10) *de novo* MVs (≥2%) not listed at P0 for the heterogeneous population (Mixed as referred in the original article), while 3 (P5) and 4 (P10) *de novo* MVs were detected in the purified population. These numbers are similar to the number of *de novo* mutations (≥2%) that we found for HSV-1 strain SC16 purified populations in both cell types. Before the high-throughput sequencing era, it was also reported that the spontaneous mutation rate in laboratory strains of HSV-2 (including the 333 strain) was 9- to 16-fold more frequent than that in HSV-1 SC16, when selecting drug-resistant mutants in cell culture [27]. These data correlate with our findings observed from the genome-wide analysis of genetic diversity, where HSV-2 populations generated a 5-fold higher number of variants than HSV-1 counterparts in Vero cells, but 10-fold higher when passaged in HaCaT cells. On the other hand, different studies describing the genetic diversity found in clinical isolates of both serotypes, by high-throughput sequencing technologies, also support that HSV-2 is more prone to generate nonconservative genetic diversity than HSV-1 also *in vivo*. Seven uncultured swab specimens of genital HSV-1 showed a total of 114 summed variants (≥2%) [41], while 10 HSV-1 clinical isolates from Finland exhibited less than 150 grouped variants, mainly localized at repeated regions [32]. However, when ten neonatal HSV-2 isolates were examined, a total of 1,821 variants was found, of which 784 were found across 71 genes [33]. The degree of coding diversity was also similar when compared to a different set of 10 adult HSV-2 isolates. Additionally, it was also reported that HSV-2 isolates generated drug-resistant mutants 30 times faster than HSV-1 clinical isolates [27]. It seems reasonable to attribute the higher mutation frequency of HSV-2 to a lower fidelity of its polymerase during viral replication. Nevertheless, HSV-1 recombinants expressing HSV-2 polymerase were reported to have similar error rates to HSV-1 parental homologs, suggesting that the polymerase would not be solely responsible for these serotype-specific differences in mutation frequency, and other viral proteins and secondary structures of the genome might contribute to explain it [56].

These experiments emphasized the critical value and usefulness of using *de novo* assembled consensus genomes generated from the actual stocks under study [48,49], as well as of using purified homogeneous viral populations from the exact parental stocks. These previously described consensus genomes for strains SC16 and 333 represented with higher accuracy the structural and genetic heterogenicity contained into each parental stock, rather than having used reference strain genomes commonly used for comparative genomics [42]. The reduction of the initial genetic variability contained into each original viral stock by plaque-picked isolation of subclones, contributed to set a more uniform baseline to identify new generated MVs during viral replication. Some genome-wide studies of HSV-1 revealed approximately 3–4% of nucleotide variation genome-wide, being reduced up to 1–2% after plaque-purification of subclones [23,47,57]. We reduced this genome-wide nucleotide variation rate from 0.47% (712 MVs / 150,000 bp) and 0.7% (1044 MVs) to an average of 0.08% (118 MVs) and 0.1% (145 MVs), between five times plaque-purified clones and their parental stocks for HSV-1 and HSV-2, respectively. Nonetheless, it has been reported that the extreme genetic bottleneck

induced by this isolation technique can cause severe attenuation of mortality in mice, even the complete loss of *in vitro* replicative capacity [46]. Characterizing the biological phenotype of these plaque-isolated subclones is essential to ensure that the genetic bottleneck exerted by the experimental approach has not altered the virus biology. The measurement of these phenotypes may include classical measures, such as plaque morphology or replication kinetics, and *in vivo* measures of pathogenesis. This connection between comparative genomic studies to the measurement of biological phenotypes is critical to integrate previous phenotypic effects of over-expression, deletion, and modifications of defined loci with the new insights from comparative genomic studies.

From a technical perspective, this study highlights the importance of ultra-deep sequencing for the identification of *de novo* genetic diversity, since greater coverage depth is directly translated into higher confidence when identifying MVs. Only a few complete rounds of viral replication occur in a single passage in cell culture, where *de novo* MVs would be represented with a very low frequency within the replicating population. If these new variants are not beneficial or are just tolerable in terms of viral fitness, they might not increase their frequency enough in the population to be detectable by standard-deep sequencing, which average coverage ranges between 100-1000X [32,33,35,41,44]. Those and other studies generally reported only MVs above 2%, accepted as the minimum frequency threshold cut-off. If so, with a 100X coverage, a 2% threshold for MVs calling would mean that only two sequencing reads were required to detect the minor variant. In order to be able to lower this threshold to 1%, we increased the coverage cut-off greater than 200X. On top of these, we implemented an additional filter to allow the call of those MVs with high frequency but below the 200X coverage threshold, proportionally increasing the coverage needed for positive filtering as lower the frequency was. There are many variant callers available for low-frequency MVs detection, which use information on basecall and read mapping quality to assess if a variant detected in a read may be due to a sequencing/mapping error or truly reflects the biological diversity of the sequenced sample. VarScan2 and LoFreq variant callers have been reported as highly indicated choices for identifying low frequency variants with high confidence from heterogenic strain mixtures of large DNA viruses [58]. The fact that we observed similar and consistent results when using different variant callers, i.e. HSV-2 exhibited greater rates of MVs accumulation than HSV-1, adds an extra degree of robustness to the confidence in our findings. These comprehensive analyses and quality controls, together with the benefits provided by an ultra-deep sequence coverage, are instrumental in confidently detecting very low MVs, thus overcoming the challenges presented by short-read sequencing of high G + C content repetitive genomes [42,59]. The integration of comparative genomic and reverse genetic approaches will improve our understanding of fundamental aspects of HSV biology, where studying the phenotypic effect of *in vitro* and *in vivo* generated variants can complement previous discoveries on gene roles, as well as explaining or predicting clinical outcomes.

The human herpesviruses literature shows how researchers have used cell lines, commonly Vero cells, to generate high titer stocks of both laboratory-adapted strains but also to amplify scarce clinical samples to decipher the *in vitro* and *in vivo* aspects of HSV biology. Most of the HSV comparative genomic studies have also used *in vitro* amplification to generate a high yield of viral genomic DNA for the preparation of high-quality sequencing libraries. Other authors pointed out that growing HSV in cell culture is clearly different from how the virus would replicate within its human host, where selective pressures and genetic bottlenecks must be substantially different between these two replicative scenarios [44,47,60]. However, here for the first time, we have identified and characterized how the genetic diversity is differentially generated between human herpesviruses when serially passaged in cell culture. The effects that

this differentially generated genetic diversity may have had on each aspect of HSV biology, as well as on the clinical outcomes of infections, is currently an active area of research [59].

Alphaherpesviruses are no longer seen as a static and homogeneous population but as such presenting *in vivo* heterogeneous diversity [41,61]. HSV-1 and HSV-2 exhibit a remarkably unequal global seroprevalence and preference for infecting different anatomical areas, where each cellular environment may have exerted differential selective pressures over viral replication, latency and reactivation. HSV-2 is more closely related to Chimpanzee herpes virus than to HSV-1. A hypothetical distant evolutionary origin between HSV-1 and HSV-2 [15], together with the unique selective pressures found at each epithelial and neuronal tissue of the oronasal and genital areas, might have contributed to their preexisting evolutionary divergence and differential genetic drift rate. A better understanding of how human herpesviruses mutate during each phase of their life cycle will provide a better knowledge on sequence determinants of virulence factors and will help to monitor resistance to anti-viral drugs.

## Materials and methods

### Cells and viruses

Vero (*Cercopithecus aethiops* kidney epithelial) cells (ATCC, CCL-81) and HaCaT (human epithelial keratinocytes) cells (Section of Virology, Department of Infectious Disease, Imperial College London [62]) were maintained at 37˚C with 5 percent $CO_2$. Cells were cultured in Dulbecco's Modified Eagle's Medium (DMEM) supplemented with 5% (v/v) Fetal Bovine Serum (FBS), 2 mM L-glutamine, and antibiotics (75 µg/ml penicillin, 75 U/ml streptomycin, and 25 µg/ml gentamycin). Cells were regularly tested for mycoplasma contaminations by standard PCR with primers Myco_Fw (GGCGAATGGCTGAGTAACACG) and Myco_Rv (CGGATA ACGCTTGCGACCTAT). HSV-1 strain SC16 and HSV-2 strain 333 original stocks were kindly provided by Dr. Helena Browne, University of Cambridge (UK). The genome sequence of plaque purified viral clones from original stocks are GenBank available under accession no. KX946970 for HSV-1 strain SC16 [49], and under accession no. LS480640 for HSV-2 strain 333 [48].

### High-throughput sequencing

High-throughput sequencing was performed in a similar manner as previously described [48,49]. Briefly, viral DNA was prepared by infection of one confluent P150-cm$^2$ plate of Vero cells (MOI = 5 PFU/cell). Cells and supernatant were collected when reaching 90–100% of cytopathic effect. Viral nucleocapsids were extracted by mechanical disruption of the cellular pellet and clarified by cellular debris after 10 min of centrifugation at 300 x *g*. Viral particles were treated with DNAse I, RNAse A, and nuclease S7 to eliminate remaining cellular DNA/ RNA, and nuclease activity was then inactivated with EDTA-EGTA. Nucleocapsids were then lysed using sodium dodecyl sulfate and Proteinase K, and viral genomic DNA was purified using phenol-chloroform-isoamyl alcohol. Potential contaminating DNA was checked by PCR against mycoplasma, prokaryotic 16S rRNA (primers 16S_Fw: CCTACGGGNBGCASCAG, and 16S_Rv: GACTACNVGGGTATCTAATCC) and eukaryotic 18S rRNA (primers 18S_Fw: GCCAGCAVCYGCGGTAAY, and 18S_Rv: CCGTCAATTHCTTYAART) [63,64]. Finally, viral DNA was tested by PCR to determine HSV-type cross-contamination (primers Up_US4 (1)_Fw: AGCGCCGTTGACTACATTCAC, and Dw_US4(1)_Rv: GCGCACCGGTGATTTAT ACCA, for HSV-1; Up_US4(2)_Fw: TCTTGAGCGCCATCGACTACG, and Dw_US4(2)_Rv: CCGCTCCATAGCTGCTGTACC, for HSV-2). An aliquot of viral genomic DNA (100 ng) was submitted to MicrobesNG, University of Birmingham (UK), to prepare barcode sequencing libraries, according to the NEBNext Ultra DNA Library Prep kit instructions (New

England Biolabs). Libraries were quantified by Qubit (Invitrogen, CA), assessed by Bioanalyzer (Agilent), and library adapter qPCR (KAPA Biosystems). Sequencing was performed on an Illumina MiSeq device as paired-end reads (2 x 250 bp), according to the manufacturer's recommendations. Sequencing statistics for every sample used in this study can be found in Sheet A in S1 Table.

## Selection of viral clones from original stocks

Five plaque-purified viral clones from both HSV-1 (SC16) and HSV-2 (333) original stocks were isolated by plaque isolation. Vero cell monolayers with 5 x $10^5$ cells/well in 6-well plates were infected at an MOI of 0.01 PFU/cell. After 48 hours post-infection (hpi), defined and single viral plaques were carefully isolated by fine pipetting 10 μl of media containing the selected plaque. Then, 30 μl of fresh media were added to each isolated plaque, followed by three rounds of freezing and thawing. An aliquot of 1 μl was three times serially diluted to infect fresh Vero cells monolayer in 6-well plates. After five subsequent rounds, one confluent P150-cm$^2$ plate of Vero cells was infected in order to produce a viral stock for sequencing and subsequent infections.

## *In vitro* generation of genetic variability experiments

Two selected plaque-purified clones from HSV-1 and HSV-2 original stocks were used to infect a P60-cm$^2$ plate of Vero and HaCaT cells separately at an MOI of 0.1 PFU/cell. After 2 hpi, the viral inoculum was removed and fresh DMEM with 2% FBS was added. Forty-eight hpi viruses were harvested by collecting both cells and supernatant, followed by three freezing and thawing cycles. Each cycle of infection and harvest was considered a passage. The harvested viruses were then used to infect the next fresh plate of Vero or HaCaT cells (estimated MOI of 0.1 PFU/cell). Each selected clone was passaged ten times in each cell line. Passages from 4$^{th}$ to 5$^{th}$ and 9$^{th}$ to 10$^{th}$ were made by infecting one confluent P150-cm$^2$ plate of corresponding cells (adjusting the MOI), in order to obtain higher yields of viral DNA for sequencing.

## Genetic variant analysis and identification of *de novo* variants

Reads from each sequenced sample were trimmed using Trimmomatic v0.36 [65], quality-filtered with PrinSeq v1.2 [66], and aligned against the reference sequence for each case, by using Bowtie 2 v2.3.4.1 [67], with default settings. Alignments were visualized using Integrative Genomics Viewer v2.8.2 [68] to detect large gaps and rearrangements. MVs present in each sequenced viral population were detected by using VarScan v2.4.3 [69], with settings intended to minimize sequencing-induced errors from the raw calling of MVs: minimum variant allele frequency ≥0.01 (1%); minimum coverage ≥20, base call quality ≥20, exclusion of variants supported on one strand by >90 percent. Detected MVs from VarScan2 calling were then annotated onto their corresponding genome to determine their mutational effects.

Additionally, alignments were optimized with Picard Tools v2.25.5 (http://broadinstitute.github.io/picard) and GATK v3.8 (https://gatk.broadinstitute.org), for automated alignment improvement, and then, MVs were analyzed by VarScan2 as describe above. For Picard Tools, we used: MarkDuplicates (Optical duplicate pixel distance 2500), AddOrReplaceReadGroups, BuildBamIndex and CreateSequenceDictionary tools, with default parameters. For GATK we used: RealignerTargetCreator and IndelRealigner tools, with default parameters.

For variant analysis using LoFreq v2.1.5 [70] we first applied to each alignment, a probabilistic realignment to correct mapping errors, followed by insertion of InDels qualities. Then,

MVs were detected with default parameters, selecting those with a frequency ≥0.01 (1%) for further analysis.

Variant calling with BAMreadCount v0.8 (https://github.com/genome/bam-readcount) was performed with default settings including: minimum base quality 20, minimum read mapping quality 20. We also implemented an exclusion filter of variants supported on one strand by >90%, as well as a Fishers' exact test to determine if the distribution of forward/reverse reads supporting a MV was significantly different of the distribution supporting the reference. Detected and filtered MVs with a frequency ≥0.01 (1%) were used for further analysis.

MVs detected by using each variant calling software were then additionally filtered by coverage >200. An additional filter was implemented in order to detect those MVs with high frequency but low coverage, where read depth at the given position had to be greater than the product obtained from dividing 200 (coverage threshold) by the variant frequency (0–100) at the given position:

$$read\ depth\ at\ the\ given\ position > \left( \frac{200\ (Coverage\ threshold)}{frequency\ (0-100)} \right)$$

MVs were considered as *de novo* appearance when, after coverage filtering, its frequency in the previous parental viral population was non-existent or < 0.01. Coverage plots for each alignment, as well as detected MVs (with VarScan2) in each viral population, were represented across their corresponding genome, according to their location and frequency (S1–S3 Figs and S5–S9 Figs). For a summary list of detected, filtered, and categorized MVs for every sample sequenced in this study, see Sheet B in S1 Table. For full lists of MVs detected in each viral population, see Sheet C in S1 Table (HSV-1 original stock and isolated clones), Sheet D in S1 Table (HSV-2 original stock and isolated clones), Sheet E in S1 Table (ultra-deep sequencing of HSV-1 clones 2 and 3), Sheet F in S1 Table (ultra-deep sequencing of HSV-2 clones 1 and 5), and Sheets G and H in S1 Table (HSV-1 clone 2 in Vero and HaCaT cells, respectively), Sheets I and J in S1 Table (HSV-1 clone 3 in Vero and HaCaT cells, respectively), Sheets K and L in S1 Table (HSV-2 clone 1 in Vero and HaCaT cells, respectively), and Sheets M and N in S1 Table (HSV-2 clone 5 in Vero and HaCaT cells, respectively). For a summary list of total and *de novo* MVs detected with each variant calling software package, for every sample sequenced in this study, see Sheets O and P in S1 Table, respectively.

## Statistical analysis

All statistical analyses were performed using GraphPad Prism 8 (v8.4.3) software. Two-tailed Mann–Whitney *U*-test was used for the number of MVs analyses ($p < 0.05$).

## Supporting information

**S1 Fig. Schematic of the HSV-1 strain SC16 (A) and HSV-2 strain 333 (B) sequenced genomes from original stocks.** Each CDS is presented in forward (red) or reverse (blue) orientation. Detected MVs (Sheets C and D in S1 Table) are mapped as black (not *de novo*) or red (*de novo*) dots across the genome, according to their location (*x*-axis) and frequency (*y*-axis). GC% plots (purple lines) and coverage plots from data alignments (blue/orange profiles) have also been mapped across each genome.
(TIF)

**S2 Fig. Variant analysis of HSV-1 plaque-purified clones.** Coverage plots from data alignments are represented in blue, for each individual case. Detected MVs (Sheet C in S1 Table) are mapped as black (not *de novo*) or red (*de novo*) dots across the genome, according to their

location (*x*-axis) and frequency (*y*-axis). MVs were considered as *de novo* when these were not previously found in the original stock (see Material and Methods for details).
(TIF)

**S3 Fig. Variant analysis of HSV-2 plaque-purified clones.** Coverage plots from data alignments are represented in orange, for each individual case. Detected MVs (Sheet D in S1 Table) are mapped as black (not *de novo*) or red (*de novo*) dots across the genome, according to their location (*x*-axis) and frequency (*y*-axis). MVs were considered as *de novo* when these were not previously found in the original stock.
(TIF)

**S4 Fig. Replication kinetics and pathogenesis of HSV-1 and HSV-2 plaque-isolated clones compared to their corresponding original stocks.** (A) Vero cells were infected with the indicated viruses at high MOI (5 PFU/cell) for one-step growth curves, and at low MOI (0.01 PFU/cell) for multi-step growth curves. Virus titers from fractions containing cell-associated virus were determined by plaque assay at 24 hpi in the one-step curves, and at the indicated times in the multi-step curves. Graphs display means and SD from two independent experiments performed in triplicate. (B) Female BALB/c mice (n = 5) were infected with the indicated virus and dose, by intranasal (i.n.) or intravaginal (i.v.) inoculations. Mice were monitored daily for survival, body weight, and signs of illness. Weight data are expressed as the mean +/- SEM of the five animal weights compared to their original weight on the day of inoculation. Signs of illness, as a score ranged from 1 to 4, is also expressed as the mean +/- SEM of the five animals. A colored "1" indicates thereafter only one animal remained in that group. Statistical analysis was performed for bodyweight data, using multiple *t*-tests with Sidak-Bonferroni correction ($p < 0.05$).
(TIF)

**S5 Fig. Variant analysis of HSV-1 plaque-purified clones 2 and 3 (A) and HSV-2 clones 1 and 5 (B) from high-depth sequencing data.** Coverage plots from alignments are represented in blue or orange, for each case. Detected MVs (Sheets E and F in S1 Table) are mapped as black (not *de novo*) or red (*de novo*) dots across the genome, according to their location (*x*-axis) and frequency (*y*-axis). MVs were considered as *de novo* when these were not previously found in the corresponding original stock.
(TIF)

**S6 Fig. Variant analysis of HSV-1 plaque-purified clone 2, after 5 and 10 passages in Vero and HaCaT cells.** Coverage plots from high-depth sequencing data alignments are represented in blue. Detected MVs (Sheets G and H in S1 Table) are mapped as black (not *de novo*) or red (*de novo*) dots across the genome, according to their location (*x*-axis) and frequency (*y*-axis). Mutations from passage 0 were considered as *de novo* when these were not previously found in the original stock, whereas those from passage 5 and 10, regarding passage 0.
(TIF)

**S7 Fig. Variant analysis of HSV-1 plaque-purified clone 3, after 5 and 10 passages in Vero and HaCaT cells.** Coverage plots from high-depth sequencing data alignments are represented in blue. Detected MVs (Sheets I and J in S1 Table) are mapped as black (not *de novo*) or red (*de novo*) dots across the genome, according to their location (*x*-axis) and frequency (*y*-axis). Mutations from passage 0 were considered as *de novo* when these were not previously found in the original stock, whereas those from passage 5 and 10, regarding passage 0.
(TIF)

**S8 Fig. Variant analysis of HSV-2 plaque-purified clone 1, after 5 and 10 passages in Vero and HaCaT cells.** Coverage plots from high-depth sequencing data alignments are represented in orange. Detected MVs (Sheets K and L in S1 Table) are mapped as black (not *de novo*) or red (*de novo*) dots across the genome, according to their location (*x*-axis) and frequency (*y*-axis). Mutations from passage 0 were considered as *de novo* when these were not previously found in the original stock, whereas those from passage 5 and 10, regarding passage 0.
(TIF)

**S9 Fig. Variant analysis of HSV-2 plaque-purified clone 5, after 5 and 10 passages in Vero and HaCaT cells.** Coverage plots from high-depth sequencing data alignments are represented in orange. Detected MVs (Sheets M and N in S1 Table) are mapped as black (not *de novo*) or red (*de novo*) dots across the genome, according to their location (*x*-axis) and frequency (*y*-axis). Mutations from passage 0 were considered as *de novo* when these were not previously found in the original stock, whereas those from passage 5 and 10, regarding passage 0.
(TIF)

**S10 Fig. Comparison of different variant calling software packages detecting preexisting (white) and *de novo* generated MVs (red) between HSV-1 or HSV-2 purified clones, after five (P5) and ten (P10) passages in Vero and HaCaT cells.** Total number of MVs are plotted according to variant analysis data (Sheets O and P in S1 Table) performed with Picard, GATK and VarScan2 (A), LoFreq (B), and BAMreadCount software (C).
(TIF)

**S1 Animation. Variant analysis of HSV-1 plaque-purified clone 2, after 5 and 10 passages in Vero and HaCaT cells.** See S6 Fig for additional details.
(GIF)

**S2 Animation. Variant analysis of HSV-1 plaque-purified clone 3, after 5 and 10 passages in Vero and HaCaT cells.** See S7 Fig for additional details.
(GIF)

**S3 Animation. Variant analysis of HSV-2 plaque-purified clone 1, after 5 and 10 passages in Vero and HaCaT cells.** See S8 Fig for additional details.
(GIF)

**S4 Animation. Variant analysis of HSV-2 plaque-purified clone 5, after 5 and 10 passages in Vero and HaCaT cells.** See S9 Fig for additional details.
(GIF)

**S1 Table. (A) Genome sequencing statistics for each sample sequenced in this study.** Legend: SRA (Sequence Read Archive), QF (quality-filtered). **(B) Categorized number of MVs for each sample sequenced in this study.** Legend: NCR (non-coding region), CR (coding region). **(C) List of detected MVs from deep sequencing of HSV-1 original stock and plaque-purified clones 1–5.** Legend: Ref (reference allele), Var (variant allele), NCR (non-coding region), INS (insertion), DEL (deletion), *name*_freq (variant allele frequency), *name*_cov (total coverage), *name*_Ref (reference allele coverage), *name*_Var (variant allele coverage). **(D) List of detected MVs from deep sequencing of HSV-2 original stock and plaque-purified clones 1–5.** Legend: Ref (reference allele), Var (variant allele), NCR (non-coding region), INS (insertion), DEL (deletion), *name*_freq (variant allele frequency), *name*_cov (total coverage), *name*_Ref (reference allele coverage), *name*_Var (variant allele coverage). **(E) List of detected MVs from ultra-deep sequencing of HSV-1 plaque-purified clones 2 and 3.** Legend: Ref (reference allele), Var (variant allele), NCR (non-coding region), INS (insertion), DEL (deletion),

*name*_freq (variant allele frequency), *name*_cov (total coverage), *name*_Ref (reference allele coverage), *name*_Var (variant allele coverage). **(F) List of detected MVs from ultra-deep sequencing of HSV-2 plaque-purified clones 1 and 5.** Legend: Ref (reference allele), Var (variant allele), NCR (non-coding region), INS (insertion), DEL (deletion), *name*_freq (variant allele frequency), *name*_cov (total coverage), *name*_Ref (reference allele coverage), *name*_Var (variant allele coverage). **(G) List of detected MVs from ultra-deep sequencing of HSV-1 clone 2 at P0, P5, and P10 in Vero cells.** Legend: Ref (reference allele), Var (variant allele), NCR (non-coding region), INS (insertion), DEL (deletion), *name*_freq (variant allele frequency), *name*_cov (total coverage), *name*_Ref (reference allele coverage), *name*_Var (variant allele coverage). **(H) List of detected MVs from ultra-deep sequencing of HSV-1 clone 2 at P0 in Vero cells, and P5 and P10 in HaCaT cells.** Legend: Ref (reference allele), Var (variant allele), NCR (non-coding region), INS (insertion), DEL (deletion), *name*_freq (variant allele frequency), *name*_cov (total coverage), *name*_Ref (reference allele coverage), *name*_Var (variant allele coverage). **(I) List of detected MVs from ultra-deep sequencing of HSV-1 clone 3 at P0, P5, and P10 in Vero cells.** Legend: Ref (reference allele), Var (variant allele), NCR (non-coding region), INS (insertion), DEL (deletion), *name*_freq (variant allele frequency), *name*_cov (total coverage), *name*_Ref (reference allele coverage), *name*_Var (variant allele coverage). **(J) List of detected MVs from ultra-deep sequencing of HSV-1 clone 3 at P0 in Vero cells, and P5 and P10 in HaCaT cells.** Legend: Ref (reference allele), Var (variant allele), NCR (non-coding region), INS (insertion), DEL (deletion), *name*_freq (variant allele frequency), *name*_cov (total coverage), *name*_Ref (reference allele coverage), *name*_Var (variant allele coverage). **(K) List of detected MVs from ultra-deep sequencing of HSV-2 clone 1 at P0, P5, and P10 in Vero cells.** Legend: Ref (reference allele), Var (variant allele), NCR (non-coding region), INS (insertion), DEL (deletion), *name*_freq (variant allele frequency), *name*_cov (total coverage), *name*_Ref (reference allele coverage), *name*_Var (variant allele coverage). **(L) List of detected MVs from ultra-deep sequencing of HSV-2 clone 1 at P0 in Vero cells, and P5 and P10 in HaCaT cells.** Legend: Ref (reference allele), Var (variant allele), NCR (non-coding region), INS (insertion), DEL (deletion), *name*_freq (variant allele frequency), *name*_cov (total coverage), *name*_Ref (reference allele coverage), *name*_Var (variant allele coverage). **(M) List of detected MVs from ultra-deep sequencing of HSV-2 clone 5 at P0, P5, and P10 in Vero cells.** Legend: Ref (reference allele), Var (variant allele), NCR (non-coding region), INS (insertion), DEL (deletion), *name*_freq (variant allele frequency), *name*_cov (total coverage), *name*_Ref (reference allele coverage), *name*_Var (variant allele coverage). **(N) List of detected MVs from ultra-deep sequencing of HSV-2 clone 5 at P0 in Vero cells, and P5 and P10 in HaCaT cells.** Legend: Ref (reference allele), Var (variant allele), NCR (non-coding region), INS (insertion), DEL (deletion), *name*_freq (variant allele frequency), *name*_cov (total coverage), *name*_Ref (reference allele coverage), *name*_Var (variant allele coverage). **(O) List of total MVs detected from variant analysis performed with four different variant calling tool sets, for each sample sequenced in this study. (P) List of *de novo* MVs detected from variant analysis performed with four different variant calling tool sets, for each sample sequenced in this study.**
(XLSX)

**S1 Text. Supporting Material and Methods for HSV replication kinetics and infection models shown in S4 Fig.** Virus growth curves protocol, ethical statement, description of procedures employed to infect and monitor mouse pathogenesis, and statistical analysis.
(DOCX)

## Acknowledgments

We thank Anthony Minson and Helena Brown (University of Cambridge, UK) for kindly providing the HSV-1 strain SC16 and HSV-2 strain 333 viral stocks, respectively. We are grateful to the Genomics and Next Generation Sequencing Service at Centro de Biología Molecular Severo Ochoa for their support and advice. Genome sequencing was provided by MicrobesNG (http://www.microbesng.uk), which is supported by the BBSRC (grant no. BB/L024209/1).

## Author Contributions

**Conceptualization:** Alberto Domingo López-Muñoz, Alberto Rastrojo, Antonio Alcamí.

**Data curation:** Alberto Domingo López-Muñoz, Alberto Rastrojo.

**Formal analysis:** Alberto Domingo López-Muñoz, Alberto Rastrojo, Rocío Martín.

**Funding acquisition:** Alberto Domingo López-Muñoz, Antonio Alcamí.

**Investigation:** Alberto Domingo López-Muñoz, Alberto Rastrojo, Rocío Martín.

**Methodology:** Alberto Domingo López-Muñoz, Alberto Rastrojo, Rocío Martín.

**Resources:** Antonio Alcamí.

**Software:** Alberto Domingo López-Muñoz, Alberto Rastrojo, Rocío Martín.

**Supervision:** Alberto Rastrojo, Antonio Alcamí.

**Validation:** Alberto Domingo López-Muñoz, Alberto Rastrojo.

**Visualization:** Alberto Domingo López-Muñoz.

**Writing – original draft:** Alberto Domingo López-Muñoz.

**Writing – review & editing:** Alberto Domingo López-Muñoz, Alberto Rastrojo, Antonio Alcamí.

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
