## [Decision Letter · Decision Letter 0]

10 May 2021

Dear Prof. Alcami,

Thank you very much for submitting your manuscript "Herpes simpex virus 2 (HSV-2) evolves faster in cell culture than HSV-1 by generating greater genetic diversity" for consideration at PLOS Pathogens. As with all papers reviewed by the journal, your manuscript was reviewed by members of the editorial board and by several independent reviewers. In light of the reviews (below this email), we would like to invite the resubmission of a significantly-revised version that takes into account the reviewers' comments.

Dear Dr. Alcami,

We received back comments from the three reviewers who agreed to evaluate your manuscript. I do want to apologize for the extended time required to get the decision back to you as it was difficult to secure reviewers. The opinions of the reviewers were supportive and they all noted the value of the work. That being said, it appears that there were some significant concerns raised by the reviewers that would need to be addressed in order to secure an acceptance for publication in PLoS Pathogens. I would consider revisiting a revised version of the manuscript with the concerns of the reviewers addressed. Their specific concerns are listed below.

Cheers,

Eain Murphy.

We cannot make any decision about publication until we have seen the revised manuscript and your response to the reviewers' comments. Your revised manuscript is also likely to be sent to reviewers for further evaluation.

Sincerely,

Eain A Murphy, Ph.D.

Associate Editor

PLOS Pathogens

Shou-Jiang Gao

Section Editor

PLOS Pathogens

Kasturi Haldar

Editor-in-Chief

PLOS Pathogens

orcid.org/0000-0001-5065-158X

Michael Malim

Editor-in-Chief

PLOS Pathogens

orcid.org/0000-0002-7699-2064

Dear Dr. Alcami,

We received back comments from the three reviewers who agreed to evaluate your manuscript. I do want to apologize for the extended time required to get the decision back to you as it was difficult to secure reviewers. The opinions of the reviewers were supportive and they all noted the value of the work. That being said, it appears that there were some significant concerns raised by the reviewers that would need to be addressed in order to secure an acceptance for publication in PLoS Pathogens. I would consider revisiting a revised version of the manuscript with the concerns of the reviewers addressed. Their specific concerns are listed below.

Cheers,

Eain Murphy.

Reviewer's Responses to Questions

**Part I - Summary**

Reviewer #1: HSV-1 and HSV-2 infections are prevalent and contribute to clinical pathology in patients worldwide. Recent studies done using high-throughput sequencing has revealed that these viruses are heterogeneous and rapidly evolve under a variety of selection pressures, including, but not limited to cell type, and understanding both the heterogeneity and the selection pressures that induce these genetic drifts is critical to understanding how these viruses could become resistant to antivirals, among other things. In this current manuscript, the authors performed standard deep sequencing and ultra deep sequencing on two commonly used viral stocks in HSV-1 and HSV-2 research labs and quantified the variants present in these stocks. Following plaque purification of clones from each stock, the authors reinvestigated variants in the clones after confirming that the virulence and in vivo phenotypes of the plaque purified clones was comparable to the original stocks. Importantly, they showed through deep sequencing of these passaged stocks that the earlier passages were more homogenous, and that with increasing passages the number of minor variants increased, and this was cell type dependent and finally HSV-2 variants were much more frequent. Overall, the manuscript is well done and the data is convincing. The authors show clearly that even after relatively low passages of viral stocks, that genetic variants rapidly appear, a finding relevant to many researchers that utilize passaged HSV-1 and HSV-2 stocks in research today.

Reviewer #2: In this article, “HSV-2 evolves faster in cell culture than HSV-1 by generating greater genetic diversity” by Lopez-Munoz, et al, the authors investigated the genetic variations in HSV-1 and HSV-2 in two cell types (Vero and HaCaT cells) over time by serial passage. The have compared deep and ultradeep sequencing of these passages over time to reveal genetic variation, including minor variants (MVs), SNPs, and InDels. Not surprisingly, the depth of sequencing increases the ability to detect genetic diversity present in low frequencies. While the sequencing data and analyses appear adequate, what is lacking is a connection to why this is important to viral fitness. While the authors performed growth curves, and one, very small in vivo experiment in mice, which are buried in the supplemental data, these experiments are just scratching the surface. As is, these data provide the foundation, but the subsequent analyses on the impact of viral infection, pathogenesis and spread are lacking. This is also important because it is not clear, at this point, whether this is simply a tissue culture phenomenon. If this data were from clinical samples over time, maybe this dataset would be ok, but it seems unsurprising that serial passage of any virus in tissue culture will result in genetic variations, if only to become more adapted to the tissue culture environment. In summary, while the sequencing data and analyses are adequate, this story is incomplete without accompanying data on how such diversity impacts the virus and/or virus-host relationship. Specifics are provided below.

Reviewer #3: López-Muñoz et al have performed a comparative study that aims to measure the rates at which HSV-1 and HSV-2 generate sequence diversity in vitro. Whether intentional or not, this study provocatively implies that HSV-1 and HSV-2 are capable of generating RNA virus-levels of diversity, a claim that has also been made for other herpesviruses (chiefly HCMV) and has also been subject to counter studies that refute these claims, most of which are due to technological/methodological artefacts. Significant care is therefore required both in the design and execution of experiments, but also in the interpretation and presentation of the data. The authors generally perform well in this regard but I do still have a few significant concerns that are detailed below and that I hope they are willing and able to address.

**Part II – Major Issues: Key Experiments Required for Acceptance**

Reviewer #1: NA

Reviewer #2: Major:

1. This story is incomplete and is highly descriptive. The importance of how quickly HSVs diversify upon serial passage in vitro is unclear.

2. HSV-1 strain SC16 and HSV-2 strain 333 were chosen. What is the rationale? Would other strains commonly used among researchers display similar genomic diversity over serial passages?

3. It is unclear if the diversity observed is simply a tissue culture phenomenon. While this might prove important and change the way in which researchers handle the virus (in other words, avoid serial passaging on Veros), it is unclear how this actually impacts viral fitness. The authors show one set of images from one clone to describe the differences in plaque formation, but how consistent is this? Is this due to the passage of the Vero cells themselves? Is this relevant to actual changes and genetic diversity in an individual? If this diversity happens much more slowly in individuals (presumably it does), then what is our take away from this tissue culture experiment beyond we shouldn’t be passaging virus ad nauseum?

4. The overall effect on viral fitness, growth, pathogenesis is not clear. There is only one experiment that assesses viral growth (at high and low MOI), yet there are no error bars. There is a single experiment in mice, using only 5 animals per infection condition. These data also reveal no statistically significant phenotype (at least none are indicated on the presented graphs). Nonetheless, this work-up is minimal, and does not provide a cohesive picture of the impact of these mutations on viral fitness, growth, or replication.

Reviewer #3: Overall, the study is robust with significant care taken to address many of the obvious concerns associated with minority variant detection. However, there are several details missing from the Methods section that I would like to see addressed along with one additional analysis.

1. One potential issue with ultra-deep sequencing is that the proportion of PCR duplicated fragments that are sequenced increases. It is therefore important to assess the level of duplicate reads and remove these prior to analysis (e.g. using picard MarkDuplicate). It is not clear to me from the methods whether the authors did this.

2. HSV-1 and HSV-2 genome sequences contain numerous homopolymers that are very challenging for short-read sequencing approaches to deal with as they are prone to PCR errors (during library preparation), and alignment errors. Local realignment strategies can help mitigate the latter and should be implemented.

3. VarScan2, while useful in many respects, relies on input data formatted using the SAMtools mpileup command. There are several issues with the generation of mpileup files, particularly when dealing with small genomes with high depths of coverage. In my experience, this leads to a vast overinflation of MVs called. I think it is thus critical that the authors perform a secondary analysis using an alternative software approach that does not rely on mpileup files. At minimum I would suggest using either LoFreq (https://csb5.github.io/lofreq/) or bamreadcount (https://github.com/genome/bam-readcount). Both are powerful, work directly from BAM files, and in the case of bamreadcount, allow for additional (custom scripted) filtering of variant calls by determining the average position of a SNV/indel within each read (e.g. one can filter out those with a strong 5’ or 3’ bias) and/or comparing average base call qualities for the parent and variant allele.

My expectation is that once these additional analyses are dealt with is that the overall results will not significantly change (i.e. HSV-2 will show greater rates of MV accumulation than HSV-1) but that the numbers of MVs that are robustly detected will be less for most/all samples. If this is not the case and the original results remain consistent then I that is also important to note (i.e. multiple distinct analysis methods generate very similar results) as it lends added weight to the conclusions.

**Part III – Minor Issues: Editorial and Data Presentation Modifications**

Reviewer #1: There are a few areas where the wording in the results is slightly confusing. Examples include line 238 (inexistent at P0 and P5, should read non-existent)

Line 245 .....even low genetically diverse purified viral populations. - Maybe should read populations with low diversity.....

Reviewer #2: Minor:

1. There are some typos/phrasing that should be corrected by editorial review.

2. Line 92: HSV-1 doesn’t always eventually lead to encephalitis. Should rephrase to “and sometimes leads to encephalitis”.

3. In the Introduction, the authors state that diversity can be generated over multiple cycles of latency and reactivation (lines 145-146). This is certainly possible even in the non-genital context. However, I don’t think this can be connected to passaging of stocks in tissue culture (as implied on the following line).

4. At first mention of HaCaT (line 181), add a brief description of what these cells are, so readers don’t have to dig in the Methods.

Reviewer #3: 1. The final statement presented in the discussion and abstract suggests that differences in the rate of MV generation in HSV-1 and HSV-2 may be due to evolutionary divergence associated with adapting to difference anatomical niches. I am not sure the evidence supports this line of speculation over other possibilities, particularly in lieu of previous studies into the origins of HSV-1 and HSV-2 e.g. https://www.ncbi.nlm.nih.gov/pmc/articles/PMC4137711/ - which should also be cited/discussed in the introduction at least.

2. Introduction – lines 138 – 150. The authors are conflating the generation of genetic diversity with evolutionary rate and also cite several papers suggesting that herpesviruses have highly diverse populations in vivo. ‘Highly’ is doing a lot of heavy lifting in this statement and the authors fail to cite any papers that counter this argument (e.g. https://www.pnas.org/content/116/12/5693), nor do they discuss the informatics challenges presented by short-read sequencing approaches, particular in regard to robust and reliable variant calling (a significant problem). More balance is thus required in this part of the text.

3. Fig. 2 - the authors present an analysis of synonymous and non-synonymous SNV (panels A & C) but this is problematic given that the full coding diversity of HSV-1 (and likely HSV-2) is far more complex than can be ascertained from the existing GenBank annotation (https://www.nature.com/articles/s41467-020-15992-5). This is also problematic when defining coding versus non-coding regions.

4. Fig. 5 - the authors refer to ‘genes’ when in fact I think they mean open reading frames (ORFs).

PLOS authors have the option to publish the peer review history of their article (what does this mean?). If published, this will include your full peer review and any attached files.

Reviewer #1: No

Reviewer #2: No

Reviewer #3: No
---

## [Editor Report · Decision Letter 1]

15 Jul 2021

Dear Prof. Alcami,

We are pleased to inform you that your manuscript 'Herpes simpex virus 2 (HSV-2) evolves faster in cell culture than HSV-1 by generating greater genetic diversity' has been provisionally accepted for publication in PLOS Pathogens.

Best regards,

Eain A Murphy, Ph.D.

Associate Editor

PLOS Pathogens

Shou-Jiang Gao

Section Editor

PLOS Pathogens

Kasturi Haldar

Editor-in-Chief

PLOS Pathogens

orcid.org/0000-0001-5065-158X

Michael Malim

Editor-in-Chief

PLOS Pathogens

orcid.org/0000-0002-7699-2064

Dear Dr. Alcami,

Thank you for resubmitting your manuscript. We do appreciate the detailed response to reviewers' criticisms. It is the opinion of the editorial staff that you have significantly improved the manuscript and we feel it is suitable for acceptance in PLOS Pathogens in its new form. Congratulations and we hope for continued success.

Cheers.

Eain Murphy.
---

## [Editor Report · Acceptance letter]

4 Aug 2021

Dear Prof. Alcamí,

We are delighted to inform you that your manuscript, "Herpes simplex virus 2 (HSV-2) evolves faster in cell culture than HSV-1 by generating greater genetic diversity," has been formally accepted for publication in PLOS Pathogens.

Best regards,

Kasturi Haldar

Editor-in-Chief

PLOS Pathogens

orcid.org/0000-0001-5065-158X

Michael Malim

Editor-in-Chief

PLOS Pathogens

orcid.org/0000-0002-7699-2064